# Ayahuasca as a Decoction Applied to Human: Analytical Methods, Pharmacology and Potential Toxic Effects

**DOI:** 10.3390/jcm11041147

**Published:** 2022-02-21

**Authors:** Ľuboš Nižnanský, Žofia Nižnanská, Roman Kuruc, Andrea Szórádová, Ján Šikuta, Anežka Zummerová

**Affiliations:** 1Department of Forensic Medicine and Toxicology, Health Care Surveillance Authority, Antolská 11, 85107 Bratislava, Slovakia; roman.kuruc77@gmail.com (R.K.); andrea.baloghova11@gmail.com (A.S.); jansikuta@me.com (J.Š.); anezka.zummerova@gmail.com (A.Z.); 2Institute of Forensic Medicine, Faculty of Medicine, Comenius University in Bratislava, Sasinková 4, 81108 Bratislava, Slovakia; 3Department of Analytical Chemistry, Faculty of Natural Sciences, Comenius University in Bratislava, Mlynská Dolina, Ilkovičova 6, 84215 Bratislava, Slovakia

**Keywords:** ayahuasca, human, pharmacokinetics, toxicity, therapeutic, analytical methods

## Abstract

Ahyahuasca is a term commonly used to describe a decoction prepared by cooking the bark or crushed stems of the liana *Banisteriopsis caapi* (contains *β*-carbolines) alone or in combination with other plants, most commonly leaves of the shrub *Psychotria viridis* (contains N,N-dimethyltryptamine-DMT). More than 100 different plants can serve as sources of β-carbolines and DMT, which are the active alkaloids of this decoction, and therefore it is important to know the most accurate composition of the decoction, especially when studying the pharmacology of this plant. The aim was to summarize the latest sensitive methods used in the analysis of the composition of the beverage itself and the analysis of various biological matrices. We compared pharmacokinetic parameters in all of the studies where decoction of ayahuasca was administered and where its composition was known, whereby minimal adverse effects were observed. The therapeutic benefit of this plant is still unclear in the scientific literature, and side effects occur probably on the basis of pre-existing psychiatric disorder. We also described toxicological risks and clinical benefits of ayahuasca intake, which meant that the concentrations of active alkaloids in the decoction or in the organism, often not determined in publications, were required for sufficient evaluation of its effect on the organism. We did not find any post-mortem study, in which the toxicological examination of biological materials together with the autopsy findings would suggest potential lethality of this plant.

## 1. Introduction

Ayahuasca is a Quechuan term used to describe *Banisteriopsis caapi* a jungle liana of the Malpighiaceae family. However, the term ayahuasca is more commonly used for psychoactive halucinogenic tea-like beverage from the Amazon and Orinoco river basins in South America [1]. The etymology of the word ayahuasca in the Quechua language of the original inhabitants of Peru and Ecuador comes from the terms “*aya*”-spirit (soul, world of death) and “*huasca*”-liana or vine, which in English can be translated as “vine of the soul” [2]. The same drink (decoction) is called “*honi zuma*” in another language used on the banks of the Rapiche River (Peru, Brazil), and in Colombia the general term “*yagé*” is used instead of ayahuasca. In the language of the Caxinaua Indians (Peru, Brazil) it is called “*honi*”; in Ecuador, specifically in Achuar and Shuar ethnic group, term “*natem*” is used. The Portuguese transcription of the word ayahuasca is *hoasca* (or *oasca*), this name is also recognized in Brazil [3]. 

Since the 1930s, in addition to shamanic consumption, ayahuasca has also been used by syncretic religious groups that have sprung up in Brazil. The first one was founded by R.I. Serra. During his stay in the jungle, he learned how to use ayahuasca from indigenous people. After returning to the civilized world, he created a new ritual way of exploiting the effects of this drink and founded the Santo Daime religion. This is a typical religious syncretism, a junction between the Christian religion and alternative healing and shamanic practices. In 1945, Barquinha separated from Santo Daime. This group created União do Vegetal (UDV), a religious society of Brasilian origin, in 1961. After the death of R.I. Serra in 1971, Santo Daime split into several fractions [3,4]. In the 80s of the 20th century, syncretistic religions spread in Brazil, and later on they established branches abroad e.g., in the Netherlands, USA, and some other countries [5]. Since the end of the twentieth century, “ayahuasca tourism” has become a cultural phenomenon. Ayahuasca has undergone a process of globalization and its use has spread throughout Asia [6,7]. In this regard, there has been a growing debate about the fact that N,N-dimethyltryptamine (DMT) is a controlled substance in many countries, which means that it is illegal to make, buy, possess, or distribute it [8]. The psychoactive effects of ayahuasca cannot be achieved without a specific combination of the main plant components. It is interesting that these effects were discovered without further research of modern science.

The decoction is traditionally prepared by cooking the bark or crushed stems of the liana *Banisteriopsis caapi* (contains *β*-carbolines) alone or in combination with other plants, most commonly leaves of the shrub *Psychotria viridis* (contains N,N-dimethyltryptamine) [9,10]. The tea itself has a bitter taste and is not pleasant to drink [11]. Thus, the main compounds of the preparation are β-carboline alkaloids (harmala alkaloids-harmine, harmaline and tetrahydroharmine) and N, N-dimethyltryptamine (DMT). Ayahuasca is unique in its pharmacological activity, which is dependent on a synergistic interaction between active alkaloids in both plants, a currently well known mechanism in which the monoamine oxidase (MAO) inhibitory action of harmala alkaloids allow the hallucinogenic effects of metabolically labile DMT [12]. β-Carbolines, which act as monoamine oxidase A (MAO-A) inhibitors, inactivate MAO-A in the gut and liver. MAO-A deamination of DMT itself makes it inactive after oral ingestion, even in amounts up to 1000 mg [13]. It becomes orally active when ingested in the presence of a MAO inhibitor such as β-carbolines. This allows DMT to cross the blood-brain barrier intactly and exert its effect in the central nervous system [14,15,16,17,18,19]. Following the administration of ayahuasca, DMT and β-carboniles are present in the human organism and are metabolized and excreted in the urine [20,21]. DMT is metabolized to the oxidative deamination product 3-indole-acetic acid obtained by the MAO pathway, and to the oxidative product DMT-N-oxide (DMT-NO) mediated by cytochrome P-450. Further N-demethylation produces N-methyltryptamine (NMT), which later turns to 2-methyl-1,2,3,4-tetrahydro-β-carboline (2-MTHBC) [19]. Harmine and harmaline are demetylated into harmol and harmalol and hydroxylized by the CYP450 enzyme complex. CYP1A2 and CYP2D6 are suggested to be major isoenzymes responsible for the catalysation. Harmol, harmalol and tetrahydroharmol (metabolite tetrahydroharmine (THH)) are conjugated to harmol sulfate (or glucuronide), harmalol sulfate (or glucuronide) and tetrahydroharmol sulfate (or glucuronide) [19]. These second-stage biotransformation reaction products have been found in human urine, as evidenced by significantly elevated urinary concentrations of harmol, harmalol and tetrahydroharmol (THHOH) after enzymatic hydrolysis using *β*-glucuronidase/sulfatase [20,21]. In humans, DMT, DMT-NO, THH, THHOH, harmine, harmol, harmaline, harmalol, 2-MTHBC, indole 3-acetic acid (3-IAA) were found in the blood after the ingestion of ayahuasca; in addition to these substances, NMT was present in the urine [20,21,22]. The hallucinogenic effect of ayahuasca is mediated through the serotonin receptor 5-HT_2_, of which DMT is an agonist (5-HT_A/2C_) [23]. However, DMT has also an affinity for other 5-HT subtypes of the receptor (1A, 1B, 1D, 2A, 2B, 2C, 6 and 7) [19]. β-carbolines as harmine and harmaline can also be hallucinogenic in sufficient quantities [24], and for their different structural analogues different affinities, mainly for serotonin receptor subtypes (5- HT1A, 2A, 2B, 2C), have been observed [19,25]. Harmine, harmaline and THH are primarily reversible inhibitors of the A-type isoenzyme of the MAO and, moreover THH exerts selective serotonin reuptake inhibitor effects [19,26]. 

The presence of endogenous DMT in the human brain has been detected, namely in the pineal gland and cerebrospinal fluid. Strassman has provided a hypothesis that DMT could be produced by the pineal gland itself, which would explain extracorporeal, mystical and spiritual experiences. DMT’s own production should increase especially in extreme situations, such as agony preceding the onset of death. A 2013 study confirmed the presence of endogenous DMT in the mammalian brain; to be specific, DMT was detected in a microdialysate obtained from the rat pineal gland. In 2019, experiments showed that the rat brain is able to synthesize and release DMT. These results confirm the hypothesis that DMT synthesis and release may similarly occur in the human brain [27].

In the presented work we focus on the administration of ayahuasca as a mixture of compounds in its entirety, not as individual substances applied to the organism, interacting in the body. We offer a review of the newest analytical methods used for the analysis of the components of ayahuasca decoction as well as their quantification in various biological matrices in humans. We summarize in detail the pharmacokinetics of ayahuasca observed in various studies with a direct link between decoction doses and corresponding blood concentrations and their physiological and psychological effects in humans. In terms of ongoing and current discussions about the therapeutic benefit and toxic risk of this plant, we also analysed general acute and chronic physiological and psychological effects, but also adverse reactions and potential toxic effects of ayahuasca in this article.

## 2. Materials and Methods

We summarized mostly (seldom, for some essential citations we also used older sources, which were never summarized) the sources published in the past 10 years focusing on the analyses of ayahuasca decoction in its entirety, not by its individual components. Books were also included in the cited literature. The following keywords were used to search for publications: ayahuasca and human and analytical methods, ayahuasca and pharmacology, ayahuasca and toxicity, ayahuasca and pharmacokinetics, ayahuasca and overdose, ayahuasca and intoxication, ayahuasca and therapeutic application, ayahuasca and post-mortem in the PubMed database (https://www.ncbi.nlm.nih.gov, accessed on 18 February 2022) and Scopus.

## 3. Physiochemical Properties of Ayahuasca Main Components and Methods of Analysis

### 3.1. Plants Used to Prepare Ayahuasca Decoction

Ayahuasca is traditionally consumed as a decoction containing an extract from the Amazonian vine *Banisteriopsis caapi*, which is rich in alkaloids, mainly in β- carboline alkaloids-harmine, harmaline and tetrahydroharmine [28]. A plant containing these substances-*Peganum harmala* (P. harmala), is used in some cases as a substitute for *B. caapi*, [29]. Some other plants containing DMT such as *Psychotria viridis* or *Mimosa tenuiflora (M. tenuiflora)* are added to the decoction to increase its psychoactive properties [18,28,30]. Depending on the origin and development of the plants used in the preparation of ayahuasca beverages, the chemical composition may vary, both qualitatively and quantitatively [10].

### 3.2. Structure and Physico-Chemical Properties of Ayahuasca Components 

β–carbolines are alkaloids derived from tryptophan, which is an aromatic amino acid containing an indole ring. L-tryptophan is formed by the shikimate pathway initially by the reaction of anthranilic acid with phosphoribozyl diphosphate and finally with L-serine. Beta-carboline alkaloids are complex indole alkaloids (a simple indole alkaloid is e.g., psilocibin and melatonin). They are formed by the reaction of tryptophan with a carbonyl compound, the carbon of which becomes part of the newly formed tricyclic structure. When the aldehyde is acetaldehyde or pyruvic acid, simple carboline alkaloids are formed and are referred to as “beta-carbolines” (in the case of acetaldehyde) or as “harmala alkaloids” (harmine, harmaline), as they were originally isolated from the plant *Peganum harmala*. They are also found, for example, in the plants of *Passiflora incarnata L.* These substances are able to reversibly block the activity of MAO enzyme subtype A [18,29]. MAO naturally degrades endogenous neurotransmitters and potentially dangerous exogenous amines that could be accidentally consumed in the diet. One of these “potentially dangerous” foreign amines is psychedelic DMT, present in large amounts in the leaves of *P. viridis* [29]. The structures of the individual active ingredients are shown in Figure 1. In the human body, these substances are metabolized and converted to their oxidized forms, DMT to indole-3-acetic acid (IAA), N, N-dimethyltryptamine-N-oxide (DMT-NO), 2-methyl-1,2,3,4-tetrahydro-β-carboline (2-MTHBC), N-methyltryptamine (NMT); harmine to harmol; harmaline to harmalol; and tetrahydroharmine (THH) to tetrahydroharmarmol (THHOH) [21]. The structure of various metabolites found in the human body is shown in Figure 2.

DMT in Ayahuasca comes mainly from *P. viridis* and its concentration ranges from 0.1% to 0.66% in dry matter [31]. β-carbolines in ayahuasca decoctions are mostly derived from *B. caapi.* These compounds represent 0.05% to 1.95% of dry matter and are much more concentrated in seeds and roots than in stems and leaves [32]. Different procedures used for the preparation of ayahuasca decoction, as well as the use of different parts of plants of the same or different species, can result in a high variability of concentrations of individual components. Active ingredients of ayahuasca are sparingly soluble in water and almost insoluble at room temperature. During the preparation of the decoction from these plants at elevated temperature, the active ingredients are partially pre-distilled into the aqueous medium. Regarding the proportional representation of the individual active ingredients of ayahuasca in decoctions, the most represented of β-carbolines is harmine, which is usually more than 15 times higher in concentration than harmaline. Different literature sources report different concentrations and proportions of active ingredients in different ayahuasca decoctions, as can be seen in Table 1.

Author Goncalves et al. compared the total phenolic compound content and flavonoid content by colorimetric method directly in the 4 most used plants for the preparation of ayahuasca decoction (*P. viridis, B. caapi, M. hostilis, P. harmala*) [34]. The results show that *M. hostilis* has the highest phenolic compound concentration (up to 376.80 ± 15.84 mg/g) and *P. harmala* has the lowest one (78.27 ± 5.75 mg/g). The highest flavonoid concentration from the examined plants is contained in *P. harmala* (25.92 ± 2.56 mg/g). In the decoction itself, the UHPLC-TOF-MS method was used to determine the content of selected phenolic substances and flavonoids in methanol extracts [34].

The loss of phenolic substances as well as of flavonoids in the decoction was significant; however, it is necessary to mention that the authors did not determine the total phenolic content and total flavonoid content in the decoction as it was in the case of plants and thus these data cannot be accurately compared. Ayahuasca decoction is prepared by boiling the plant material in water, which in the case of sparingly soluble substances present in plants causes a low content of these substances in the decoction. In addition, flavonoids are thermolabile molecules that can decompose during the cooking process.

Most studies of ayahuasca decoction focus mainly on the composition of organic molecules. Though, in addition to organic substances present in ayahuasca decoctions or directly in the plants, the information about the composition of inorganic components is of essential importance too (Table 2). Since the decoction is applied orally, it is no less important to know the recommended daily allowances for these substances, or their maximum allowable concentrations.

The most monitored key analytes in the human body are the active ingredients of ayahuasca such as DMT, THH, harmine and harmaline. A summary of results of scientific publications since 2011 concerning the consumption of ayahuasca decoctions and their concentration determination in the monitored biological matrices, including blood (plasma and serum), urine, hair, saliva and sweat, is given in Table 3. Most research is based on the use of liquid chromatography (LC) in combination with mass spectrometry (MS) or tandem mass spectrometry (MS/MS) to provide information about the concentrations found in biological matrices of ayahuasca users.

Crucial is the information about the concentration of metabolites of active ayahuasca ingredients detected in biological samples, the level of which in the organism proves the application of the decoction. In the case of DMT, plasma concentrations of its metabolites are higher than the concentrations of the active substance alone [20,39]. From active ingredients-DMT, harmine, harmaline and THH, THH is the most represented one and its levels in plasma are up to twice as high as the concentration of other active ingredients. The least represented component in plasma is harmaline [20,37,38,39]. To determine the active ingredients of ayahuasca and their metabolites in blood, authors McIlhenny et al. developed a LC-MS/MS method in 2011, which has a narrower calibration range than the more recently published LC methods, but is applicable for a wider range of metabolites. The authors used a Zorbax Eclipse HT C_18_ (1.8 µm × 4.6 × 50 mm) column and a gradient elution with mobile phases composed of 0.1% formic acid in H_2_O and 0.1% formic acid in acetonitrile for separation [20]. The applicability of their analytical method is evidenced in multiple publications, which prove the utility of this method to determine ayahuasca components and their metabolites in blood.

In urine the most abundant component is also THH, and that is at the level of 49–85% of the 4 active ingredients of ayahuasca, and harmaline is the least represented substance at the level of 4–7%. In a study by Riba et al. the metabolite IAA was in urine 37 times more represented than DMT alone [21]. Another metabolite-DMT-NO had also 25 times higher urinary concentration than the parent drug (DMT) [21]. By enzyme urine treatment, it is possible to increase analytic yield (limit of detection (LOD) and limit of quantitation (LOQ)) and to determine analytes that are not visible (quantifiable) without this step due to instrumental and methodological limitations [21]. Biological matrices such as sweat, hair and saliva are alternative samples for determining an individual’s exposure to certain drugs or other chemicals. Their advantage is non-invasiveness, painless collection compared to blood collection, but due to the smaller number of published data, especially in terms of assessing the effects of physiological and psychological status, they still remain only alternative matrices in practice [42,45]. In the case of sweat and hair, the preference lies in the possibility of detecting certain drugs and medicines even after a longer period of time, in contrast to blood [46,47]. On the other hand, there are limitations, including intra and intervariability among individuals, as well as a lack of data about the possible contamination [48]. Table 3 lists the publications where the authors detected the ayahuasca components in hair, saliva and sweat. In the hair, the authors focused only on the determination of DMT [42,43]. In sweat and saliva also other active ingredients of ayahuasca as THH, harmine and harmaline were analysed in addition to DMT [40,44]. However, there is a lack of more extensive studies where the authors would monitor metabolites of these active substances in alternative biological matrices.

## 4. Pharmacology of Ayahuasca Main Components 

The pharmacology of the substances applied orally in the form of ayahuasca decoction is different from the pharmacology when administered as DMT, harmaline, harmine and THH in another form. The pharmacology of the decoction is affected by the ratio of individual components found in the plants used for its preparation. Therefore, in each study we focused comprehensively on thorough information about the respondents, accurate definition of the composition of the decoction (or applied dose), indication of the maximum achieved plasma concentration (C_max_), the time to maximum plasma concentration (T_max_), the elimination half-life (T_1/2_), and last but not least the description of the observed effects of the decoction. 

### 4.1. Pharmacokinetics Parameters of Ayahuasca 

Specific concentration of ayahuasca active ingredients in human organism after the ingestion of this decoction was first published in 1996 [49]. The only four published human studies [39,40,50,51] to date, evaluating the pharmacokinetic parameters (C_max_, T_max_, T_1/2_) of ayahuasca decoction components after its drinking, are summarized in Table 4. The Table also shows the concentrations of ayahuasca alkaloids (DMT, harmine, harmaline, THH, NMT, 5-OH-DMT, harmol, harmalol and THH-OH) in the decoction and their recalculated doses (mg and mg/kg of body weight). 

Average oral doses for DMT, harmine, harmaline and THH were 55.95 mg (0.79 mg/kg_of body weight (BW)_), 178.9 (2.5 mg/kg_BW_), 22.4 (0.3 mg/kg_BW_) and 161.4 (2.1 mg/kg), respectively [39,50,51]. At these doses, average maximum plasma concentrations (C_maxAVG_) for DMT, harmine, harmaline and THH were reached with values of 17.76 ng/mL, 112.43 ng/mL, 6.9 ng/mL and 120.56 ng/mL, respectively [39,50,51]. C_maxAVG_ were achieved at an average time T_max_ = 1.8 h for DMT, 2.1 h for harmine, 2.1 h for harmaline and 2.9 h for THH [39,50,51]. The average T_1/2_ for DMT, harmine, harmaline and THH was 2.2 h. The baseline data for the calculation of individual pharmacokinetic parameters and their ranges are shown in Table 4. 

With increasing DMT dose is also increasing its C_max_ (T_max_ = 1.8 h). In terms of increasing dose and C_max_, a similar trend was observed for harmaline and THH. For harmine, this trend could not be assessed due to insufficient data (Table 4). The highest C_maxAVG_ = 112.43 and 120.56 ng/mL were for harmine and THH, respectively. As for harmine, its concentration in decoction was high as well as its applied dose. C_maxAVG_ 120.56 ng/mL for THH was the highest from all, and was reached at the longest average T_max_ = 2.7 h. THH persisted in the blood for the longest time, which indicates more than three times longer T_1/2_ = 6.12 h. This, together with its high C_max_, may be partly due to the fact that the THH concentration in the beverage, as well as its average applied dose of 161.4 mg (2.1 mg/kg), was one of the highest. C_maxAVG_ = 6.94 ng/mL for harmaline was the lowest, which correlates with its low beverage concentration and low average applied dose. DMT, harmine and harmaline reached their maximum plasma concentrations at approximately the same time. DMT (T_1/2_ = 2.2 h) and harmine (T_1/2_ = 1.9 h) remained in the blood for approximately the same length of time. 

The overall evaluation of all so far published pharmacokinetic data has its limits. Table 4 shows, for example, the problem of the analysis of ayahusca alkaloids in plasma [39,50,51] and serum [40]. Specifically for DMT, in a study by Lanaro et al. ([40], Table 4) at a relatively high average dose of DMT 233 mg (3.11 mg/kg), serum C_max_ was only 7.69 ng/mL. A similar trend can be seen from the table for the C_max_ of harmine, harmaline and THH. One possible explanation is that the serum would not necessarily contain proteins involved in blood clotting to which the main constituents of ayahuasca could be bound. However, we cannot scientifically substantiate this claim, because the literature does not describe the relevant studies. Another limitation is that the average T_1/2_ for harmine and harmaline could only be calculated from two studies. In the study of Callaway et al., harmaline was detected in only 6 out of 15 respondents, despite the good sensitivity of the analytical method (LOD = 0.05 ng/mL) [49]. This suggests that the absence of harmaline in the blood is probably the result of the intervariability of the respondents. As can be seen in the table below, at similar as well as at lower concentrations of harmaline observed in other studies, its maximum concentration was detected. Surprisingly in the study of Riba et al., harmine was detected in only 4 out of 18 respondents [51], which was insufficient for the assessment of pharmacokinetic parameters (C_max_, T_max_ and T_1/2_). 

Although the study by Lanaro et al. [40] appears to have a limitation in the analysis of serum ayahuasca components compared to other plasma studies, this is the first study focusing on the kinetics of ayahuasca alkaloids in saliva [40]. C_max_ (ng/mL) was highest in saliva for THH 337.4, then for DMT 98.36, harmine 43.06 and harmaline 33.12, whereby T_max_ = 0.5 h was achieved for all the components at the same time. Half-life (T_1/2_) was 4.8 h for THH and was the highest from all of the analysed substances in the saliva. Excluding harmine, serum DMT, harmaline and THH levels were always lower than the concentrations in saliva. The results of the study also show an increasing ratio of DMT, harmine, harmaline and THH concentrations in blood to their salivary concentration, which is in correlation with mean residence time (MRT). MRT is higher for these substances in blood and correlates with the fact that the concentrations of these substances in serum, excluding harmaline, decrease more slowly than in saliva. Harmaline concentrations in the blood and serum decreased equally rapidly [40]. 

#### 4.1.1. Pharmacokinetics of Ayahuasca Metabolites 

In addition to early mentioned and most frequently studied substances, such as DMT, harmine, harmaline and THH, which were generally analysed in terms of their amount in plants, decoction or their pharmacokinetic parameters, some metabolites of DMT such as IAA, NMT, DMT- NO, 5-OH-DMT and metabolites of harmala alkaloids such as harmalol, harmol, THHOH were investigated by Schenberg et al. [39] (Table 4). NMT, 5-OH DMT, harmol, harmalol and THHOH have been detected in the ayahuasca decoction. The highest maximum plasma concentration was for IAA (C_max_ = 717.72 ng/mL), while the lowest belongs to NMT (C_max_ = 3.10 ng/mL) [43]. Contrary to other studies summarized in Table 4, the sampling interval of this study was only up to 3.5 h after the ingestion of the decoction and the elimination half-life was not evaluated. T_max_ for all above mentioned metabolites was in range 1.85–2.47 h. There are no more studies to compare C_max_ and T_max_ with these values. Pharmacokinetics of metabolites harmol and harmalol was investigated in two studies [39,51] (Table 4) with significantly different C_max_, probably due to their different concentrations in the decoction. Harmalol had a significantly higher T_1/2_ [51] (Table 4), which could be due to the metabolism of harmaline to harmalol rather than to harmine and that of harmol [19]. Approximately two times higher C_max_ of harmol than harmalol may in turn be due to 15 times higher concentration of harmine than harmaline in the ayahuasca decoction (Table 4).

#### 4.1.2. Amount of Ayahuasca Alkaloids and Their Metabolites in the Urine

Riba et al. performed an analysis of alkaloids and their metabolites in urine. The concentration in the beverage (neither initial decoction volume nor the weight after lyophilization was stated) is very likely the same as in the study by Riba et al. [51] (Table 4) and McIlhenny et al. [20]. As the weight of the respondents of this study was known (67.0 kg, range 60–85), the whole-body dose was calculated to 67 mg (60–85) of DMT, 113.9 mg (102–144.5) of harmine, 7.71 mg (6.9–9.78) of harmaline, 91.12 mg (81.6–115.6) of THH, 2.412 mg (2.16–3.06) of harmol, and 0.563 (0.504–0.714) of harmalol. Urine was collected at 0–4 h, 4–8 h, 8–16 h, and 16–24 h intervals and analyzed with and without *β* -glucuronidase/sulfatase enzyme hydrolysis. A certain basal amount of IAA was also found in the urine before the administration. IAA is the major metabolite excreted in the urine after the administration of DMT (oxidative deamination product obtained from the MAO pathway) [19]. After the consumption of ayahuasca, the amount of IAA was highest at all time intervals (up to 25,563.3 µg in 24 h). This is in correlation with highest C_max_ in the plasma [39]. DMT-NO was present in the urine at the second highest amount in all time intervals (7002.5 µg in 24 h). The third highest concentration (484.1 µg in 24 h) was attributed to DMT. It can be observed that during the first 8 h, approximately 95% of DMT, 66% of IAA, 88% of DMT-NO, 58% of 2-MTHBC, and 73% of NMT were excreted in the urine of their total excretion in 24 h. Regarding the residual amount of DMT and its metabolites from the originally applied dose of 67 mg DMT as the parent drug, the unchanged DMT excreted in urine represented 0.8%, IAA 44.2%, DMT-NO 10.2%, 2-MTHBC 0.16%, and NMT 0.03% (total 55.4%). Regarding the urinary excretion of harmala alkaloids, THH was excreted in the urine in the largest amount at all time intervals (5969.5 µg in 24 h). Its metabolite tetrahydroharmol had the second highest amount (1704.4 μg), followed by harmalol (1006.9 µg), harmol (600.8 µg), and harmaline (40.0µg) in 24-h urine. The lowest concentration belonged to harmine with a total concentration of 40.0 µg in 24-h urine. Concerning the excretion of harmala alkaloids detected in urine after hydrolysis, the amounts of harmine, harmol, harmalol and tedrahydroharmol increased significantly in contrast to harmaline and THH [21]. 

### 4.2. Common Psychological and Physiological Effect of Ayahuasca in Human

Ayahuasca is a mixture of several active ingredients, the ratio of which in the final drink is very variable ([10], Table 4), and therefore its effects depend on both the specific plants used and the method of preparation. The effects are remarkable on a physiological and mental level. However, the dosage of ayahuasca is extremely individual. It depends on the tolerance of the individual and his personal sensitivity, on the intention with which it is to be used, and above all on the intuition of the presiding mestre who works with the substance. 

#### 4.2.1. Acute Effects after Ayahuasca Administration

In a study using a standard dose of ayahuasca having the same composition as in study by Riba [51], when DMT dose was 0.5, 0.75 and 12 mg/kg, but using different respondents (71.5 kg, range 66–85), dose-dependent effects were observed. Initial somatic manifestations began to appear after 15 to 30 min after the administration of the drink, psychological manifestations after 30 to 60 min. Both somatic and psychological effects of ayahuasca reach a peak between 60 and 120 min. Then they gradually decrease and disappear completely about 240 min after the ingestion. Physically, ayahuasca has a purging effect, vomiting and diarrhea are frequently present [52]. 

In a study by Schenberg et al., vomiting was present in all but one respondent. A positive correlation between vomiting and T_max_ was significant only for harmaline, THH and IAA. No correlation between vomiting and C_max_ or AUC (area under the curve) was observed [39]. In the initial phase, *ayahuasca* can increase pulse rate (on average of 5–6 pulses per minute), blood pressure, and slightly also respiratory rate. Other manifestations may include buzzing or ringing in the ears, changes in thermoregulation in the form of hot or cold flashes spreading throughout the body, sweating, tremor, dilation of the pupils, dizziness, headaches, drowsiness, yawning, or paresthesia (tingling, itching, prickling etc.). Coordination of body movements may also be impaired [27,50,51,53,54,55]. In a study by Callaway, the enlargement of the pupil diameter started after 40 min and persisted even after 6 h and the respiration rate was slightly increased to its highest value after 90 min with fluctuations to its maximum value after 6 h. This correlates with the elimination half-life of T_1/2_ = 4.32 ± 3.45 h. The cardiovascular effect consisted in an increase of heart rate (maximum after 20 min) and increase of blood pressure (maximum after 40 min), which then decreased very slightly to normal values. Neuroendocrine effect of ayahuasca was found in a significant increase of growth hormone, prolactin and cortisol to their peak plasma concentrations occurring between 1–2 h after application [50]. In a study by Riba, regarding the effects on the cardiovascular system, only diastolic blood pressure was significantly changed at 75 min after the administration of the beverage [51]. 

Harmine and harmaline are competitive and reversible inhibitors of monoamine oxidase (MAOI), while THH is thought to inhibit serotonin reuptake [19]. Harmine also plays a role in bone and cartilage regeneration [56,57], and acts as a potent antibiotic [58] and antiparasitic in the body [59]. Equally important is the impact of harmine on the nervous system as it has sedative, anxiolytic and antidepressant effects [53,60,61]. To a small extent, β-carboline alkaloids themselves are also psychoactive on their own if their dosage is sufficient. They cause visual hallucinations and states of euphoria [24,51]. Nevertheless, *Psychotria viridis* with the DMT component plays a major role in the psychoactive properties of ayahuasca. DMT is structurally similar to the serotonin neurotransmitter and binds to serotonine receptors (5-HT_2_) types 2A, 2C and 1A in the central nervous system, where it acts as an agonist [4,62,63,64]. Lower Ki (inhibitory constant) means the higher affinity of the substance for the receptor. For 5-HT_2A_ receptors, DMT (Ki = 323 nM) has a comparable affinity for this receptor as harmine (Ki = 397 nM), harmaline has an affinity approximately 16-fold lesser and THH has a much lower affinity (Ki > 10,000 nM). For 5-HT_2C_ receptors, DMT (Ki = 1450 nM) has a higher affinity than harmine (Ki = 5340 nM), harmaline (Ki = 9430 nM) and THH (Ki > 10000 nM). A similar trend was observed for 5-HT_1A_ receptors [62].

Various positive psychological effects were reported; many respondents described delightful feelings after the application of ayahuasca. All the respondents felt stimulated in some way [51]. Immediate psychological effects include transient perceptual changes-hallucinations, changes in sensory perception, especially visual [65]. Synesthesia, changes in perception of time and space, changes in cognitive functions in general, as well as intensified emotional experience are often described by ayahuasca users. Under the influence of ayahuasca, the mind may be capable of deeper introspection and various memories arise [54]. In clinical trials, transient perceptual, cognitive and affective modifications (typical for psychedelics) are associated with physiological symptoms such as increased systolic and diastolic blood pressure, increase of cortisol and prolactin, lymphocyte redistribution, or electroencephalographic changes [39,51,66].

#### 4.2.2. Late Effects after Ayahuasca Administration

Scientific research focusing on the long-term effects of hallucinogens are rare. In 2013, Guimarães dos Santos, based on the available scientific literature, stated that long-term members of Brazilian religious groups with prolonged ayahuasca intake (some up to 10 years) show good tolerance. No physiological toxicity or psychological, neuropsychological or psychiatric harm has been observed [67]. Later, the long-term effects of ayahuasca in naive and chronic users were also examined, and it was found that both groups had better scores for depression after half a year [68] or psychometric improvements were sustained [69]. Approximately four weeks after the ayahuasca ritual, participants were found to have improved cognitive thinking style and mindfulness [70]. Some of the above studies admit that psychotic-like adverse reactions or, in some individual cases, anxiety [68] may occur, which would help to clarify further clinical studies.

## 5. Ayahuasca Toxicity and Toxicological Risks and Hazards to Human Health

### 5.1. Common Side Effects of Ayahuasca

As stated in the pharmacological characteristics of ayahuasca beverage, its toxicity should be different from the toxicity of its individual ingredients if administered separately. Therefore, in this section we will focus only on the analysis of the toxicity, side effects and possible risks of ayahuasca beverage in its entirety. 

Some rare case reports suggest that ayahuasca administration may have played a role in isolated deaths, but the lack of key information (such as past medical history, blood analysis, drink composition, the amount of ingested dose) gives significant limits to draw conclusions about the direct causal relationship. Information reported in media (TV, newspapers) also suggest the existence of significant, sometimes fatal adverse reactions, but such reports do not provide conclusive forensic evidence, such as autopsy findings or the chemical composition of the ingested material. 

Nausea, vomiting and diarrhea may be considered to be relatively common acute physical adverse reactions following the administration of ayahuasca [39,43,50,51,55]. In some cases, however, vomiting is not observed despite high concentrations of individual Ayahuasca alkaloids in the decoction [40]. In the case of harmine, nausea, vomiting, tremor and numbness of the body were reported, and it was associated mainly with intravenous administration of the drug or after high oral doses. Harmaline and harmine usually caused unpleasant physical sensations, especially paresthesia and numbness, intense vomiting and dizziness. Focusing problems with eyes were also observed [24,71]. A direct relationship between the level of harmaline in the beverage and the degree of vomiting was described [39]. It also triggered the urge to defecate, especially at the onset of effects. Vomiting and diarrhea may be considered to be the result of increased serotonin levels in the gastrointestinal tract [11]. Callaway [50] described that vomiting was induced by increased vagus nerve stimulation at serotonin receptors in the central region. Elevated stimulation of serotonin receptors in the peripheral region in turn increased intestinal motility, resulting in diarrhea. From the aspect of potential health hazards, especially vomiting is considered dangerous, because it can theoretically lead to aspiration of gastric contents and thus endanger health and even life. Despite this, generally many users do not consider vomiting or diarrhea to be a discomfort, but rather a kind of purge effect for the body [55]. Ingestion of higher doses may be accompanied by an increase in blood pressure and pulse rate, which may lead to dangerous side effects in case of pre-existing heart disease. No adverse effects were observed in long-term ayahuasca users or in elderly people [7]. From a medical point of view, a combination of ayahuasca with prescription drugs, especially SSRI antidepressants (selective serotonin reuptake inhibitors), means a great risk. Their combination can cause serotonin syndrome, which is a potentially life-threatening condition [6,52,55,72,73]. It is certainly worth mentioning that high doses of manganese were found during the analysis of the inorganic composition of ayahuasca (Table 2). Mn exceeds the recommended daily dose by more than 6 times. Manganese (Mn) is an essential trace element that is part of several enzymes in the body. However, in cases of overexposure, whether inhaled or ingested, Mn is highly toxic to several organ systems. Mn crosses the blood-brain barrier by the same mechanism as iron (Fe) [74,75]. Excessive exposure to Mn initially causes non-specific symptoms such as headache, irritability, fatigue, sleep disorders and emotional instability. Later, a neurodegenerative syndrome with psychiatric symptoms, called manganism, may develop, which is characterized not only by problems with speech, gait and balance, but also with obsessive-compulsive behavior, hostility, mood swings, psychotic experiences such as hallucinations and paranoid ideas, and reduced cognitive flexibility [74,76,77]. Some ayahuasca decoctions may even contain heavy metals, such as lead or cadmium, which are not recommended at all to be consumed.

Adverse reactions or even toxic effects of ayahuasca are currently studied mostly in relation to central nervous system damage or fetal development during pregnancy or in terms of intake of ayahuasca by children. One of the many statements about this topic [78,79] is given in the study by Charles Grob (co-author of important studies on ayahuasca), who alleged that the adolescents coming from União do Vegetal (an ayahuasca-based religion), who came into contact with ayahuasca during their intrauterine development, were in very good psychological health, had lower rates of anxiety, mood disorders or alcohol use disorders. 

### 5.2. Toxic Clinical Manifestations Linked to Ayahuasca (Surviving Patients)

In cases of overdose survivors, the diagnosis of poisoning should be ideally confirmed by ascertaining the specific concentration of the poison in the blood. The quantification of a particular poison is regarded as a clear proof of its presence in the organism and the only way to assign the symptoms of poisoning to a given substance. This is often misunderstood (even in the case of ayahuasca) and blood tests are replaced by the examinations done on urine or stomach contents, whereby the presence of a toxic substance in stomach contents does not necessarily mean its toxic concentration in the bloodstream. However, the blood analysis on suspicion of toxicity emergencies is difficult in practice, both because of relative short half-life of its components and the fact that not all hospital laboratories have the capacity to analyze those compounds. The chance to find compounds from ayahuasca in urine is much higher even if they do not reflect blood levels. This together with the symptoms should be sufficient for diagnostic purposes. Frison et al. described a case of β-carbonyl intoxication when ingesting *Peganum harmala* seed extract [24]. This plant can be used as a β-carbonyl component together with another DMT-containing component for the preparation of ayahuasca. However, no traces of DMT, 5-methoxy-DMT or hallucinogen tropane alkaloids were detected in this case. On the other hand, harmine 60,000 ng/mL, harmaline 450,000 ng/mL and THH (not quantified) were found in the urine. These concentrations significantly exceeded the concentrations of harmine 160 ng/mL and harmaline 510 ng/mL in 0–4 h time interval, concentrations of harmine 120 ng/mL and harmaline 500 ng/mL in 4–8 h interval and concentrations of harmine 10 ng/mL and harmaline 320 ng/mL in 8–24 h interval in urine samples (volunteers) collected after drinking ayahuasca decoction [22]. A concentration of harmine 7200 µg/mL and harmaline 12 000 µg/mL was found in the seed extract, which was much higher than in commonly prepared ayahuasca drink (Table 4) or when using *Banisteriopsis caapi* [10] to prepare the drink (harmine 109–7110 µg/mL, harmaline 12–945 µg/mL). The patient’s symptoms were consistent with the symptoms of intoxication by harmala alkaloids (psychomotor agitation, visual and auditory hallucinations, diffuse tremors, locomotor ataxia, nausea, vomiting etc.). A neurological examination showed that he was unable to stand upright, sleepy, though responding to verbal stimulus, and presented intention tremor and tremor in sustained posture (upper and lower limbs). Unfortunately, no toxicological analysis of the blood was performed [24]. This was also the case of another 17-year-old boy who was admitted to hospital after the ingestion of an extract from three Syrian rue seeds; he smoked 10 mg of 5-MeO-DMT and snorted an additional 15–20 mg of 5-MeO-DMT. These substances are remarkably similar to the combination of β-carbonyls with tryptamines as in ayahuasca. The symptoms after the ingestion of the extract were extremely combative; the patient was unable to answer questions, there was severe agitation, pupils were dilated, and the skin was diaphoretic. He experienced one bout of emesis on the route to the hospital. Concentrations of these substances in the blood and the urine were not determined, only harmine and harmaline were confirmed qualitatively by GC-MS method in urine [80]. 

Regarding central nervous system, ayahuasca toxicity may be associated with psychiatric disorders. Such a case was observed in a man with bipolar disorder, which appeared two days after his participation in the ayahuasca ritual. Manic symptoms such as mystical and paranoid delusional ideas, auditory hallucinations, racing thoughts, disorganized behavior, elevated energy, and euphoria, were manifested in him. Psychotic symptoms were consistent with his euphoric mood. The manic symptoms were concluded to be the result of the antidepressant effect of ayahuasca (similarly to other antidepressants). This may be an example of another mechanism of ayahuasca toxicity in bipolar patients [81]. It is well known that ayahuasca may have antidepressant effects (reviewed in [82]) and also that the development of manic episodes in psychiatric patients after the administration of antidepressants is a criterion for the diagnosis of bipolar disorder [81]. However, in this study, no active alkaloids found in ayahuasca or other antidepressants were analysed in the organism (blood or urine).

Another study analysed the case of a 25-year-old man with a history of schizophrenia and suicide attempts, who was found on the street disturbing the peace, shouting and having aggressive behavior. He had mild clonus with 3+ patellar reflexes, dilated pupils, and reddened skin without diaphoresis. The patient confessed that he had drunk a tea ordered online containing ayahuasca. During the day, his condition returned to baseline. Laboratory analysis of the urine was done, whereby urinary DMT was >2000 ng/mL. These concentrations significantly exceeded DMT concentration 450 ng/mL in interval 0–4 h, DMT concentration 600 ng/mL in interval 4–8 h and DMT concentration of 30 ng/mL in interval 8–24 h in the urine samples (volunteers) collected after drinking ayahuasca beverage [22]. Examination of DMT in blood or other toxicological screening examinations in blood were not performed. However, this patient had acute psychosis resulting in self-harming on the basis of schizophrenia, which is a rare case described in the literature [73].

### 5.3. Post-Mortem Diagnosis

As it is observed in cases of various different toxicologically active substances, the toxicity of ayahuasca may be affected by the presence of another substance used simultaneously or some time before or after the administration of ayahuasca. First, it is necessary to emphasize that we have not found any confirmatory post-mortem studies that would clearly demonstrate that the cause of death was directly related to the application of ayahuasca beverage. There is a demand for such studies to be realized, especially regarding comprehensive description of autopsy findings, thorough toxicological analysis (especially the determination of the concentration of individual substances in the body), precise assessment of the cause of death (the basic cause of death and the cause directly leading to death), and eventual toxic effects on organs caused by the ingestion of ayahuasca. 

We have examined one post-mortem analysis that was associated with 5-MeO-DMT, found in some ayahuasca drinks [83]. Sklerov et al. [84] reported concentrations of individual alkaloids found in ayahuasca beverage, namely DMT, 5-MeO-DMT, tetrahydroharmine, harmaline, and harmine. The highest concentration (ng/mL) in the blood drawn from peripheral venipuncture was for 5-MeO-DMT 1200, followed by tetrahydroharmine 240, harmine 80, harmaline 40, and DMT 10. The highest concentration (ng/mL) in the blood taken from the heart was for 5-MeO-DMT 1880, THH 380, harmine 170, harmaline 70, and DMT 20. In post-mortem interpretation of the results, it is recommended to assess the concentrations of substances in the blood taken from the peripheral area rather than central, due to post-mortem redistribution [85]. It can also be seen in this case, when the concentrations of individual substances were different in blood drawn from different places. For DMT, the concentration of 10 ng/mL met the range of mean maximum concentrations in experimental volunteers or UDV members (12–25 ng/mL), where no significant side effects other than vomiting were observed (citations in Table 4). THH concentration of 240 ng/mL, except for one study (328 ng/mL), significantly exceeded the maximum mean THH concentrations (23, 44, 39, 91 ng/mL) listed in Table 4. The concentration (ng/mL) of harmaline 40 significantly exceeded the concentrations of harmaline in comparison with previous studies (1.2–15), and the concentration of harmine (80) fell within the concentration range (16–114) as in the studies presented in Table 4. The concentration of 5-MeO-DMT in blood (1200 ng/mL) and in the stomach content (201,000 ng/mL) was also high [84] in contrast to the concentration of other components found in the body after drinking Ayahuasca beverage. This most likely indicates acute poisoning by this substance. Although 5-MeO-DMT is generally described to be a component of ayahuasca [83], its concentration detected in the studied case was markedly high, also when compared to different ayahuasca beverages [86]. Besides of its high concentration in the blood and the stomach content, the concentration of this substance was also significantly high in the urine (9591 ng/mL). In similar manner as DMT, 5-MeO DMT is detected in higher concentrations in blood when co-administered with harmala alkaloids [87]. Since the concentrations of 5-MeO-DMT, and also of harmaline, in the blood significantly exceeded the concentrations of these substance detected in humans in various clinical studies and taking into account a high oral dose (gastric content), and accumulation of this substance in liver, kidney and brain, it was ascertained that the cause of death was intoxication by 5-MeO-DMT [84]. It is in correlation with the animal study, in which persistent high accumulation of 5-Me-O-DMT was observed in liver, kidneys and brain [87]. If 5-Me-O-DMT can be detected in ayahuasca [83] along with THH, harmine, harmaline and DMT, then the participation of ayahuasca drink in the death of this patient cannot be excluded.

## 6. Potential Benefits of Ayahuasca

Ayahuasca was and is considered by Native Americans to be the most powerful medicinal plant on Earth. They attribute the ability to teach and heal people to the decoction made out of this “magical” liana, and some even believe that it mediates the connection with the universe and spiritual beings. In recent decades, the use of ayahuasca has spread from South America not only to Europe and to the United States, but throughout the whole world, and ayahuasca has become the subject of various biomedical studies. These studies and research have raised hopes for its therapeutic potential, but also concerns about its possible toxicity [6,55,88]. Authors Galvão Ana et al. emphasized in their work the importance of the effect of ayahuasca on the production of salivary cortisol, which acts in the regulation of various physiological, cognitive and emotional pathways. Their opinion was based on studies that suggested that regulating salivary cortisol levels to normal values was considered an important part of the treatment of depression. It should be further explored with the emphasis on the need for clinical trials with natural psychedelics applied in cases of mental disorders [89]. Some studies suggest that the main active ingredients of ayahuasca (DMT and β-carbolines) have anti-inflammatory, neuroprotective and memory-improving effects [90]. Galvão-Coelho et al. observed a reduction in C-reactive protein (CRP) levels 48 h after the intake in ayahuasca-treated patients but not in placebo-treated subjects. However, the exact mechanism is unknown [91]. The use of ayahuasca during the day subsequently affects the sleep cycle; however, does not worsen the quality of sleep. A prolongation of the second sleep stage, shortening of the duration of rapid eye movement (REM) sleep phases and prolongation of non-REM phases was described [92]. Multiple studies suggest that the use of ayahuasca may be beneficial for people with mood disorders, depression, anxiety or post-traumatic stress disorder [61,70]. At the same time, the studies focused on the possibilities of its use in the fight against various addictions (alcoholism, nicotinism, cocainism etc.) [4,26,65,93]. Some authors assume that the use of ayahuasca could be implemented into the controlled treatment with health benefits [9,39,53,94]. Recent studies suggest that regular intake of psychedelic microdoses leads to positive effects on health, mood, cognitive function, and reduced depression and anxiety [9,39,53,94,95]. 

Yet DMT-containing ayahuasca appears to be less toxic while retaining psychological effects. Based on the studies of the health status of ayahuasca users, the use of ayahuasca may be considered safe and even beneficial to health [89]. Adverse results have been reported extremely rarely and are considered to be the result of uncontrolled intake of non-traditional ayahuasca preparations. However, there is still a need for more extensive clinical research on the use of these substances. Such studies should be done by recognized, credible researchers and must include a comprehensive recording of side effects as well as beneficial effects. These studies should be registered with the appropriate global clinical databases [31]. 

## 7. Conclusions and Perspectives

There are many ongoing discussions about therapeutic benefits and toxicological risks of ayahuasca plant, the outcome of which is ambiguous and still debatable at this time. However, it can be observed that the “weighing pan” is more inclined to the side of potential benefits, which is evidenced by observing long-term positive effects after the application of this plant. However, the number of publications focused on the investigation of toxicological risks of ayahuasca is not small. In particular, the limits of the relevant evaluation of its benefits and risks lie in an insufficient number of studies dealing with the analysis of the concentrations of individual components in the organism after the ingestion of ayahuasca decoction with their subsequent correlation to the effects of the plant. Last but not least, thorough investigation of pathological changes on tissues and organs in cases of long-term use of ayahuasca is also insufficient and thus requirable.

## Figures and Tables

**Figure 1 jcm-11-01147-f001:**
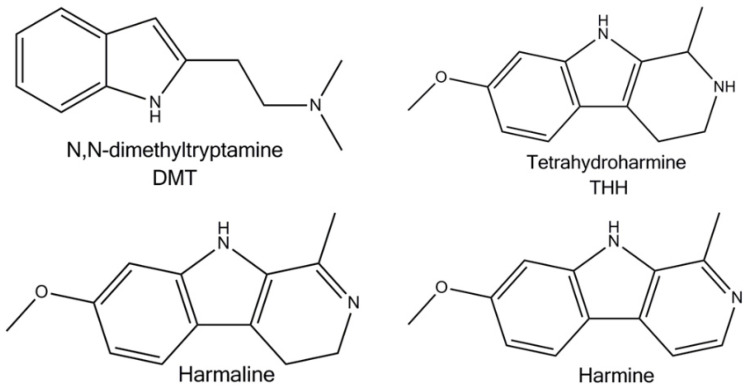
Psychedelic DMT and β-carbolines present in ayahuasca.

**Figure 2 jcm-11-01147-f002:**
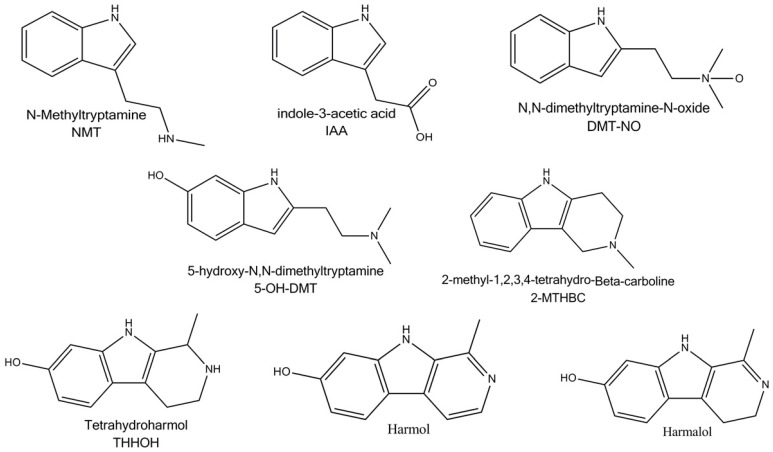
Metabolites of Ayahuasca.

**Table 1 jcm-11-01147-t001:** Psychedelic DMT and β-carbolines and their metabolites as components found in traditional Ayahuasca and in the human body. The data in the table are collected from the literature [19,29,33].

Compound	Summary Formula	M_w_ (g/mol)	Concentration in Decoction (mg/mL)	Percentual Concentration in Decoction (%)	Derivates Found in Human Body
Harmine	C_13_H_12_N_2_O	212.25	0.11–7.11	40 (15–75)	harmol
Harmaline	C_13_H_14_N_2_O	214.26	0.004–0.945	2 (0.3–16.8)	harmalol
Tetrahydroharmine	C_13_H_16_N_2_O	216.28	0.032–3.88	30 (1.4–55.4)	tetrahydroharmol
N,N-dimethyltryptamine	C_12_H_16_N_2_	188.27	0.088–3.12	25 (9.2–63,7)	DMT-NO, NMT, 2-MTHBC, IAA

IAA-indole 3-acetic acid, DMT-NO-N,N-dimethyltryptamine-N-oxide, 2-MTHBC-2-methyl-1,2,3,4-tetrahydro-β-carboline, NMT-N-methyltryptamine.

**Table 2 jcm-11-01147-t002:** Inorganic composition of ayahuasca decoctions. Daily values were calculated as a percentage of the maximum concentration in the decoction and recommended dose [35,36].

Element	Concentration Range (mg/L) in Ayahuasca Decoctions	Recommended Dose (mg/day)	Maximum Concentration in 150 mL Decoction (mg/day)	Daily Values (%)
Ca	102–664	1300	100	8
Mg	313–1542	420	231	50
P	47–616	1250	92	7
K	2017–7263	4700	1089	23
Li	0.0045–0.076	1	0.011	1
Al	<LOQ-9.7	10	1.5	15
Mn	4.8–94	2.3	14.1	613
Fe	1.75–6.9	18	1.04	6
Cu	<LOQ-0.17	0.9	0.026	3
Co	<LOQ-0.17	0.05	0.026	52
Zn	0.62–19.3	11	2.9	26
As and Hg	<LOQ	-	0	-
Cd	0.0038–0.027	-	0.004	-
Pb	<LOQ-0.35	-	0.053	-

<LOQ under quantification limit, - not applicable.

**Table 3 jcm-11-01147-t003:** Summary of publications covering consumption of ayahuasca decoction and concentration of individual substances derived from the decoction and their metabolites in human biological samples.

Analytical Method	Matrix	Analyte and Concentration [ng/mL]	Calibration Range (ng/mL)	Year	Reference
LC-MS/MS	plasma	DMT [ND-15.1] DMT-NO [0.25–45.2] THH [25.9–55.4] THHOH [0.96–2.9] Harmine [0.54–5.2] Harmol [0.32–5.55] Harmaline [2.2–4.5] Harmalol [1.7–3.3] 2-MTHBC [ND-0.48] IAA [4.3–207.8]	1–50 1–50 1–50 1–50 1–50 1–50 1–50 1–50 1–50 10–500	2012	[20]
UHPLC-MS/MS	plasma	DMT [ND-4.3] THH [13.4–117.9] Harmaline [<LOQ-2.1] Harmine [<LOQ-40.6]	1–150 2–150 1–150 1–150	2021	[37]
LC-MS/MS	plasma	DMT [1.2–19.8] Harmine [1–15.6] Harmaline [2.7–15.7] THH [27.1–71.4]	0.5–100	2012	[38]
LC-MS/MS	plasma	DMT [2.5–65.9] NMT [0.57–7.6] DMT-NO [1.6–40.6] Harmine [5.9–511.5] Harmol [0–318.8] Harmaline [1.56–44.4] Harmalol [3.1–48.6] THH [22.6–895.7] THHOH [0–111] IAA [61.35–1674.6]	1–50 1–50 1–50 1–50 1–50 1–50	2015	[39]
			1–50 1–50 1–50 10–500		
LC-MS/MS	serum	DMT [0.03–17.8] Harmine [0.1–61] Harmaline [0.1–5.4] THH [0.5–93.3]	0.3–200	2021	[40]
LC-MS/MS	serum	DMT [3.2] Harmine [12.3] Harmaline [0.3] THH [182]	0.3–200	2017	[41]
LC-MS/MS	urine	DMT [488] Harmine [60.6] Harmaline [49.4] THH [567]	5–200	2017	[41]
HPLC-MS/MS	urine	DMT [0.26–154.5] IAA [1152–5724] DMT-NO [155.2–2366] 2-MTHBC [8.08–19.4] NMT [0.52–6.12] Harmine [3.65–8.66] Harmaline [31.86–113.75] THH [307.7–1372.3] Harmol [56.2–146.1] Harmalol [121.9–193.4] THHOH [241.4–327.2]	5–200	2012	[21]
HPLC-MS/MS	urine	DMT [30–450] DMT-NO [1270–11,060] 2-MTHBC [10–130] 5-OH-DMT [140–160] Harmine [10–160] Harmaline [20–510] Harmol [40–3090] Harmalol [1250–4040] THH [10–6270]	5–200	2011	[25]
UHPLC-MS/MS	hair	DMT [5600]	30–10,000	2014	[42]
LC-MS/MS	Hair	DMT [3–1109]	3–500	2021	[43]
GC-MS	sweat	DMT [53–186.8 ng/patch] Harmine [319.8–1461.5 ng/patch] Harmaline [87.6–201.9 ng/patch]	20–1500 ng/patch *	2021	[44]
LC-MS/MS	saliva	DMT [0.07–327.2] Harmine [0.01–95.2] Harmaline [0.08–131.3] THH [0.14–623.6]	0.3–100	2021	[40]

* ng/patch is not possible to convert to ng/mL; ND-not detected; <LOQ-under quantification limit; HPLC-High-performance liquid chromatography; UHPLC-ultra high-performance liquid chromatography; GC-gas chromatography.

**Table 4 jcm-11-01147-t004:** Human pharmacokinetics of ayahuasca.

Hoasca Alkaloids (µg/mL)	Oral Dose (mg)	Volunteers (kg)	Average of Oral Dose (mg/kg)	C_max_ (ng/mL)	T_max_ (h)	T_1/2_ (h)	Ref.
DMT (240)	35.5 ± 5.3	15 male members of UDV. Used hoasca at least 10 years. Weight 74 ± 11.3 kg (chronic administration)	0.48	15.8 ± 4.4	1.8 ± 0.54	4.32 ± 3.45	[50]
harmine (1700)	252.3 ± 38.4		3.41	114.6 ± 61.7	1.7 ± 0.97	1.9 ± 1	
harmaline (200)	29.7 ± 4.5		0.40	6.3 ± 3.1	2.4 ± 1.1	-	
THH (1070)	158.8 ±24.2		2.15	91.0 ± 22	2.9 ± 0.66	8.9 ± 4.8	
DMT (530)	39.8 (30.4–47.9) 57.4 (43.7–67.7)	15 males and 3 females without psychedelic-related disorders. Weight 66.47 (50.7–79.5) (except two, without experience with ayahuasca)	0.60 0.86	12.4 (9.09) 17.44 (10.49)	1.5 (1–2.5) 1.5 (1–4)	1.07 (0.58) 1.06 (0.77)	[51]
harmine (900)	67.4 (51.6–81.2) 95.8 (74.2–114.8)		1.01 1.44	0 0	0 0	0 0	
harmaline (60)	4.6 (3.5–5.5) 6.5 (5.0–7.8)		0.07 0.1	2.48 (1.28) 4.32 (2.43)	1.5 (1–3) 2 (1–4)	2.01 (0.56) 1.95 (0.81)	
THH (720)	54.2 (41.5–65.3) 77.0 (59.6–92.3)		0.82 1.16	23.06 (11.45) 39.4 (20.63)	2.5 (1.5–3) 3 (1.5–6)	4.78 (3.95) 4.68 (1.52)	
harmol (NA)	-			10.9 (6.04) 17.57 (7.72)	1.5 (1–2.5) 2 (1–3)	1.64 (0.29) 1.49 (0.28)	
harmalol (NA)	-			6.74 (3.52) 9.59 (4.17)	2.5 (1–4) 2.75 (1.5–4)	30.33 (20.53) 48.64 (77.09)	
DMT (328)	91.13 (64.67–114.73)	12 males and 8 females, with previous experience drinking ayahuasca. Weight 74.9 ± 7.7. Partially chronic	1.21	25.39 ± 16.78	2.21 ± 0.83	-	
harmine (1080)	300.06 (221.93–377.78)		4.01	110.26 ± 137.85	2.5 ± 0.87	-	
harmaline (176)	48.9 (34.70–61.56)		0.65	14.68 ± 13.93	2.68 ± 0.78	-	
THH (1280)	355.63 (252.36–447.74)		4.75	328.76 ± 324.86	3.18 ± 0.71	-	[39]
NMT (17)	4.72 (3.35–5.95)		0.06	3.10 ± 2.12	2.03 ± 0.79	-	
5-OH-DMT (2)	0.56 (0.39–0.70)		0.007	-	-	-	
Harmol (528)	146.70 (104.10–184.69)		1.96	84.99 ± 93.57	2.47 ± 0.92	-	
Harmalol (12)	3.33 (2.37–4.20)		0.04	17.22 ± 13.64	2.08 ± 0.65	-	
THH-OH (104)	28.90 (20.50–36.38)		0.39	22.58 ± 26.76	1.99 ± 0.90	-	
DMT-NO (ND)	-			14.42 ± 12.50	1.85 ± 0.63	-	
IAA (ND)	-			717.72 ± 454.50	2.92 ± 0.66	-	
DMT (2070)	155.25–310.5	14 participants (LDV), gender was not mentioned. Used Hoasca at least 1 year. Weight 50–100 (kg), chronic	3.11	7.69 * 98.36 ^+^	1.3 * 0.5 ^+^	3.51 * 0.85 ^+^	[40]
Harmine (2894)	217.25–434.1		4.34	16.05 * 43.06 ^+^	1.0 * 0.5 ^+^	2.12 * 2.4 ^+^	
Harmaline (147.5)	11.05–22.1		0.22	1.21 * 33.12 ^+^	1.0 * 0.5 ^+^	3.29 * 3.7 ^+^	
THH (1843)	142–284		2.84	44.31 * 337.4 ^+^	2.0 * 0.5 ^+^	5.54 * 4.8 ^+^	

NA-not applicable; ND- not detected; C_max_- maximum plasma concentration; T_max_-the time to maximum plasma concentration; **T_1/2_**-the elimination half-life; * means maximum serum concentration, ^+^ means maximum saliva concentration.

## Data Availability

No new data were created or analyzed in this study.

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
