# Peer review of "Ayahuasca as a Decoction Applied to Human: Analytical Methods, Pharmacology and Potential Toxic Effects"

_jcm, 2022, doi:10.3390/jcm11041147_

Round 1
Reviewer 1 Report
This work by Niznansky et al. covers a very interesting and timely topic on the high diversitiy of ayahausca brews. The review covers apects of ayahuasca that are often not the main focus of research covering this field. However, there are major language deficits in this manuscript which make the reading very hard and the logic uncomprehensive.
Author Response
Reviewer #1:
This work by Niznansky et al. covers a very interesting and timely topic on the high diversity of Ayahausca brews. The review covers aspects of ayahuasca that are often not the main focus of research covering this field. However, there are major language deficits in this manuscript which make the reading very hard and the logic uncomprehensive.
Response: We have reviewed the manuscript for proper language usage and has made necessary changes. If this is not sufficient, we would use MDPI English language editing. Some chapters have been rewritten and their order logically changed according to the specific reviewer's requirements. We hope that the changes we have made will improve the readability of the manuscript. The changes are highlighted in yellow in the manuscript.
Yours sincerely,
Žofia Nižnanská, Lubos Niznansky and co-authors
Reviewer 2 Report
Review for Journal of Clinical Medicine, title: Spirited away: Potential therapeutic
- A brief summary
The present work is a review of sources published in the last 10 years on the composition of ayahuasca, determination of its active components in biological matrices, as well as its effects, therapeutic application and toxicity. The strength of the paper is the comprehensive compilation of relevant references for many aspects of ayahuasca properties and use together in one article.
- General concept comments
- Article: The topic is relevant and the references appropriate. Its main weakness is its lack of synthesis and evaluation of the knowledge as a whole. The paper presents in extreme detail the findings in the papers they have identified, making the reading very heavy and far too long, and is in the other hand too scarce in giving the whole picture and the general lines one expect form a review. See reference nr 18 as an excellent example of review on the same topic. Therapeutic benefits and toxicological perils are the weakest parts of this review. A list of abbreviations would be appreciated
- Specific comments
- Title: The title does not reflect the focus of the paper and should be reformulated. The aims stated in lines 106 to 116 reflect the real content of the article in a better way.
- Abstract: the abstract reflects fairly the content and is well presented.
- Line 24-25: I cannot see that the findings presented can support a general statement that side effects occur mainly on preexisting psychiatric disorder
- 1. Introduction: The introduction is thorough and adequate.
- Line 80-81: CYP450 is a complex including all the enzymes mentioned on line 81 (CYP2D6 is missing the last number). Consider rewording to “Harmine and harmaline are demetylated into harmol and harmalol and hydroxylized by The CYP450 enzyme complex. CYP1A2 and CYP2D6 are suggested to be major isoenzymes responsible for the catalysation”
- Line 91-95: the serotonin receptor is referred sometimes as 5-hydroxytryptamine-receptor, but most times as serotonin-receptor though the paper. No need to use both.
- Line 99-104 on the affinity to serotonin receptors does not belong to an introduction and it may be unnecessary unless it can be used to explain effects and in the case may be mentioned under pharmacology.
- 2. Materials and methods: OK
- 3. Physiochemical properties…and methods of analysis: the section gives a useful overview of the psychoactive components in ayahuasca. Table 1 is clear and useful. The section on inorganic composition is not useful, as the authors do not relate the theoretical toxicity potential of manganese to the observed toxic effects of ayahuasca in the section 5, toxicity.
- Table 3 needs some horizontal lines to be readable. Consider reordering columns 1 method, 2 matrix, 3 analyte, 4 concentration, 5 range, 6 year, 7 reference. Analyte and concentration together.
- Line 224: “the levels in plasma are of 58-71%” this is not a usual way to express concentration levels in blood at all. It can be used in urine when expressing the total excretion of metabolites from the same mother compound, but it is surely not used for plasma levels of different substances. Use twice as high or similar expression as you do on line 246 and onward. A figure with the excretion curve from one of the references had been useful.
- Line 260: Saliva is widely used for forensic purposes independently of the way of intake of the drug. I do not understand the sentence and do not find the reference. Check and rephrase.
- 4. Pharmacology of ayahuasca main components: this section is a reproduces the pharmacokinetic data by previous works in a very detailed manner, but fails to give an overview. What is the range of concentrations across investigations? What is the range of T ½ ? what is the variation of T max? What influences variations in these parameters? The methods section states that the paper includes sources published in the last 10 years, but many of the references used in this section are older (54-57).
- Section 4.1: includes both pharmacokinetics and pharmacodynamics. It extends to almost 6 pages and should be revised entirely, shortened substantially, and pharmacodynamics separated and moved to section 4,2.
- Line 284: I do not find reference 54 in the table.
- Line 311: Is the above from ref 55 or 54?
- Line 313: Is that an effect of CYP2D6 polymorphism?
- Section 4.2: The psychological effects should appear together under section 4.2.1
- Line 537: How can one take into account endogenous levels when drinking ayahuasca? The discussion of endogenous levels of DMT should be moved to a more general part of the paper, i.e. the introduction. It has no relevance for the effects of exogenous DMT.
- Section 4.2.1: This section gives a good and concise summary
- Line 556: Unclear how the term “detoxifying” is used here
- Section 4.2.2: The section is based in only one (or two? The placement of the citations numbers are confusing) human study and few animal studies. The quality of the one/two human studies should be better discussed.
- Line 584: Generally, rat doses can not be compared to human doses in terms of effects. How are these doses determined?
- Line 595: DMT and tolerance is an interesting point, because it affects the potencial for inducing drug dependence for hallucinogens. It should be given more place in a more general section
- Section 4.3: Therapeutic application is better placed after toxicity because it is the balance between effect and toxicity that defines its utility as a therapeutic agent. The quality and extend of the science behind the possible therapeutic use should be the focus of the section. The section seems to have a positive bias for the therapeutic use that is not supported by the investigations cited.
- Line 629-30: Public health benefits? In basis on what?
- Section 5.
- Line 676. The sentence about growing debate on DMT legal status seems out of place
- Line 683: again, is the dose given to mice representative?
- Section 5.2: The advice of blood analysis on suspicion of toxicity emergencies is difficult in practice, both because of relative short half-life of its components and the fact that not all hospital laboratories have the capacity to analyze for those compounds. The chance to find compounds form ayahuasca in urine is much higher even if they do not reflect blood levels. This together with the symptoms should be sufficient for diagnose.
- Line 708 and 758: Here, urine concentrations in one intoxication case are being compared to the concentrations in “mean urine volume excreted in 24 hours” in a metabolism study. The basis for this comparison is unclear and probably fallacious. Are the concentrations corrected for creatinine? Are the concentrations of the study a median of several concentrations calculated separately for several voids? A urine concentration from a short time observation may be naturally very different from the 24 hours total mean. This will not necessarily imply different doses.
- Line 738: “Ayahuasca toxicity is associated with psychiatric disorders” this general assessment is not substantiated in the text.
- Section 5.3:
- Line 795: concentrations in stomach content and in blood are not correlated, it cannot be unusually high with respect to stomach content, or is there something wrong with the sentence?
- Section 6: The large number of publications is not a criterion for therapeutic benefits. The size of the effects and the quality of the studies are.
Author Response
Reviewer #2:
The present work is a review of sources published in the last 10 years on the composition of ayahuasca, determination of its active components in biological matrices, as well as its effects, therapeutic application and toxicity. The strength of the paper is the comprehensive compilation of relevant references for many aspects of ayahuasca properties and use together in one article.
- Article: The topic is relevant and the references appropriate. Its main weakness is its lack of synthesis and evaluation of the knowledge as a whole. The paper presents in extreme detail the findings in the papers they have identified, making the reading very heavy and far too long, and is in the other hand too scarce in giving the whole picture and the general lines one expect form a review. See reference nr 18 as an excellent example of review on the same topic. Therapeutic benefits and toxicological perils are the weakest parts of this review. A list of abbreviations would be appreciated.
Response: The manuscript has been shortened and some parts have been rewritten for better comprehensibility as suggested by the reviewers. Abbreviations are better explained at the beginning of each chapter, as required by the Journal of Clinical Medicine instructions. Changes related to specific comments have been highlighted in yellow in the manuscript. We have not used tracking changes because manuscript would be confusing and unreadable then.
Specific comments
Title: The title does not reflect the focus of the paper and should be reformulated. The aims stated in lines 106 to 116 reflect the real content of the article in a better way.
Response: The title of the manuscript has been reformulated with regard to the aims stated in 106-116.
Abstract: the abstract reflects fairly the content and is well presented.
- Line 24-25: I cannot see that the findings presented can support a general statement that side effects occur mainly on preexisting psychiatric disorder
Response: “Mainly” has been replaced by “probably”, so that the meaning of the sentence would not be misleading.
Introduction: The introduction is thorough and adequate.
- Line 80-81: CYP450 is a complex including all the enzymes mentioned on line 81 (CYP2D6 is missing the last number). Consider rewording to “Harmine and harmaline are demetylated into harmol and harmalol and hydroxylized by The CYP450 enzyme complex. CYP1A2 and CYP2D6 are suggested to be major isoenzymes responsible for the catalysation”
Response: The sentence has been reformulated and divided according to the reviewer's suggestion.
- Line 91-95: the serotonin receptor is referred sometimes as 5-hydroxytryptamine-receptor, but most times as serotonin-receptor though the paper. No need to use both.
Response: Throughout the whole manuscript, the word 5-hydroxytryptamine-receptor has been replaced by the word serotonin receptor.
- Line 99-104 on the affinity to serotonin receptors does not belong to an introduction and it may be unnecessary unless it can be used to explain effects and in the case may be mentioned under pharmacology.
Response: The above paragraphs have been moved to the section 4.2.
Materials and methods: OK
Physiochemical properties…and methods of analysis: the section gives a useful overview of the psychoactive components in ayahuasca. Table 1 is clear and useful. The section on inorganic composition is not useful, as the authors do not relate the theoretical toxicity potential of manganese to the observed toxic effects of ayahuasca in the section 5, toxicity.
Response: The table describing the inorganic composition of the decoction has stayed in the text. We think that the table is relevant and its content could be useful for the reader. We have moved the section about the potential toxicity of manganese to Chapter 5.1.
- Table 3 needs some horizontal lines to be readable. Consider reordering columns 1 method, 2 matrix, 3 analyte, 4 concentration, 5 range, 6 year, 7 reference. Analyte and concentration together.
Response: Table 3 has been redesigned according to reviewer’s requirements.
- Line 224: “the levels in plasma are of 58-71%” this is not a usual way to express concentration levels in blood at all. It can be used in urine when expressing the total excretion of metabolites from the same mother compound, but it is surely not used for plasma levels of different substances. Use twice as high or similar expression as you do on line 246 and onward. A figure with the excretion curve from one of the references had been useful.
Response: The above sentence and units have been modified as suggested by the reviewer. Only Schrenberg 2015 has curves of components in blood dependent on time (only up to 200 min). Excretion is not apparent from the short time interval. We can see higher concentrations of THH in the table 4.
- Line 260: Saliva is widely used for forensic purposes independently of the way of intake of the drug. I do not understand the sentence and do not find the reference. Check and rephrase.
Response: Thank you for this comment. The sentence did not make sense from a forensic point of view and has been deleted. It was probably a writing mistake.
- Pharmacology of ayahuasca main components:this section is a reproduces the pharmacokinetic data by previous works in a very detailed manner, but fails to give an overview. What is the range of concentrations across investigations? What is the range of T ½ ? what is the variation of T max? What influences variations in these parameters? The methods section states that the paper includes sources published in the last 10 years, but many of the references used in this section are older (54-57).
Response: Thank you for your suggestions for this section of the manuscript. Concentrations, elimination half-lifes and Tmax have been summarized and discussed. In the methods section we have added information regarding the use of older publications. Mainly in section 4 we have used essential information gained from older publications, because many of these findings have not been reported in more recent scientific articles. This is the reason why we have used information from older publications in this manuscript.
- Section 4.1: includes both pharmacokinetics and pharmacodynamics. It extends to almost 6 pages and should be revised entirely, shortened substantially, and pharmacodynamics separated and moved to section 4,2.
Response: Section 4.1 has been revised entirely. The chapter has been shortened and the section about pharmacodynamics has been moved to section 4.2.
- Line 284: I do not find reference 54 in the table.
Response: The pharmacokinetic data in the reference 54 (Callaway, 1996) are the same as in the reference 55 (Callaway, 1999) in the table. The constituents of the ayahuasca drink in blood were initially analyzed in 1996, then published and analyzed in detail along with pharmacodynamic effects in 1999. Therefore, only the reference 55 is included in the table. However, this section has been greatly simplified and rewritten. We hope that it is now clearer.
- Line 311: Is the above from ref 55 or 54?
Response: It is from the reference 55. The explanation is in the previous paragraph.
- Line 313: Is that an effect of CYP2D6 polymorphism?
Response: Thank you for your remark and the sentence has been deleted from the text. We find it true that the polymorphism of the gene for the CYP2D6 enzyme has not been investigated by molecular biological methods in the mentioned article (Calaway 2005). Poor and fast metabolizers were distinguished only according to Cmax of harmine. Therefore, we have decided to delete that sentence.
- Section 4.2: The psychological effects should appear together under section 4.2.1
Response: Thank you for your suggestion, a relevant part of the section has been moved to the recommended part of the manuscript.
- Line 537: How can one take into account endogenous levels when drinking ayahuasca? The discussion of endogenous levels of DMT should be moved to a more general part of the paper, i.e. the introduction. It has no relevance for the effects of exogenous DMT.
Response: We agree with the reviewerʼs comment. The contribution of endogenous DMT after drinking ayahuasca decoction cannot be taken into account and the sentences have been deleted. Further information regarding endogenous DMT has been moved to the introduction section.
Section 4.2.1: This section gives a good and concise summary
Response: As recommended by the reviewer for section 4.2, we have additionally moved the pharmacodynamic effects of ayahuasca to this section.
- Line 556: Unclear how the term “detoxifying” is used here
Response: The word “detoxifying” has been changed to “purging”, which might clarify the meaning of the sentence.
- Section 4.2.2: The section is based in only one (or two? The placement of the citations numbers are confusing) human study and few animal studies. The quality of the one/two human studies should be better discussed.
Response: This section has been modified as suggested by the reviewer. Human studies have been better evaluated and this section has been expanded to include some more human studies. The studies cited were from authors who are well recognized in the field of ayahuasca topic. We have analyzed and took into account the significance of the results from animal studies and eventually decided to remove them from the text because the entire review/manuscript and its title refer to studies with human subjects.
- Line 584: Generally, rat doses can not be compared to human doses in terms of effects. How are these doses determined?
Response: There are generally accepted rules, especially for therapies, which adapt the dose studied in animals to the dose that will be studied in preclinical and clinical trials of drug introduction. However, animal studies have been removed from the manuscript because the entire review and its title refer to studies with human subjects. This section has been enlarged and includes only human studies.
- Line 595: DMT and tolerance is an interesting point, because it affects the potencial for inducing drug dependence for hallucinogens. It should be given more place in a more general section
Response: The sentence 595 was based on a study on an animal model. If the reviewer agrees, we will delete this sentence for the reason stated in the previous paragraph.
- Section 4.3: Therapeutic application is better placed after toxicity because it is the balance between effect and toxicity that defines its utility as a therapeutic agent. The quality and extend of the science behind the possible therapeutic use should be the focus of the section. The section seems to have a positive bias for the therapeutic use that is not supported by the investigations cited.
Response: The section therapeutic application has been moved and now follows the toxicity chapter. The title of this section has been changed to “potential benefits of ayahusca” according to the comments from the reviewer. We agree that using this term is more appropriate. A few relevant sentences have been added to the end of this section to clarify the section.
- Line 629-30: Public health benefits? In basis on what?
Response: We have rewritten the sentence and ,,public,, has been deleted, because we agree that the word was not used properly.
- Section 5. Line 676. The sentence about growing debate on DMT legal status seems out of place
Response: The sentence has been moved to the introduction part.
- Line 683: again, is the dose given to mice representative?
Response: As we mentioned above, there are generally accepted rules, especially for therapies, which adapt the dose studied in animals to the dose that will be studied in preclinical and clinical trials of drug development. However, as answered above, animal studies have been removed from the manuscript because the entire review and its title refer to studies with human subjects.
- Section 5.2: The advice of blood analysis on suspicion of toxicity emergencies is difficult in practice, both because of relative short half-life of its components and the fact that not all hospital laboratories have the capacity to analyze for those compounds. The chance to find compounds form ayahuasca in urine is much higher even if they do not reflect blood levels. This together with the symptoms should be sufficient for diagnose.
Response: We agree with the reviewer’s comment and we would like to thank the reviewer for this advice. Part of this information have been added and clarified in the section.
- Line 708 and 758: Here, urine concentrations in one intoxication case are being compared to the concentrations in “mean urine volume excreted in 24 hours” in a metabolism study. The basis for this comparison is unclear and probably fallacious. Are the concentrations corrected for creatinine? Are the concentrations of the study a median of several concentrations calculated separately for several voids? A urine concentration from a short time observation may be naturally very different from the 24 hours total mean. This will not necessarily imply different doses.
Response: We would like to thank the reviewer for this important comment. The total concentrations from urine excreted for 24 h (Riba 2012) have been replaced by the concentrations from urine collected at different time intervals (McIlhenny 2010) after the ingestion of the ayahuasca drink. It is presumable that in the study by Frison (2008), urinary concentrations of harmine and harmaline were at one of the time intervals (within 24 h after the ingestion) the same as in the McLhenny 2010 study. Moreover, these concentrations were extremely high (Frison, 2008). Similar reasoning could be applied to DMT (Bilhimer 2018), where we used the same study for the comparison (McIlhenny 2010).
- Line 738: “Ayahuasca toxicity is associated with psychiatric disorders” this general assessment is not substantiated in the text.
Response: We have edited the sentence and the word ,,may be,, has been added to the sentence, so that it makes the statement more hypothetical and thus precise.
- Section 5.3: Line 795: concentrations in stomach content and in blood are not correlated, it cannot be unusually high with respect to stomach content, or is there something wrong with the sentence?
Response: We thank the reviewer for this comment and we also agree that there is no correlation between stomach and blood concentrations. The sentence has been reformulated.
- Section 6: The large number of publications is not a criterion for therapeutic benefits. The size of the effects and the quality of the studies are.
Response: The sentence has been reformulated, so that the information is not misleading.
Yours sincerely,
Žofia Nižnanská, Luboš Nižnanský and co-authors
Round 2
Reviewer 2 Report
Good answers! Much better readability.